# Histone Acetyltransferases and Stem Cell Identity

**DOI:** 10.3390/cancers13102407

**Published:** 2021-05-17

**Authors:** Ruicen He, Arthur Dantas, Karl Riabowol

**Affiliations:** 1Arnie Charbonneau Cancer Institute, Cumming School of Medicine, University of Calgary, Calgary, AB T2N 4N1, Canada; ruicen@ualberta.ca (R.H.); arthur.dantas1@ucalgary.ca (A.D.); 2Department of Molecular Genetics, Temerty School of Medicine, University of Toronto, Toronto, ON M5S 1A8, Canada; 3Department of Biochemistry and Molecular Biology, Cumming School of Medicine, University of Calgary, Calgary, AB T2N 4N1, Canada

**Keywords:** chromatin, histone acetyl transferases, epigenetic, development, stem cells, cancer, senescence

## Abstract

**Simple Summary:**

Gene regulation is the major mechanism that allows us to develop from a single cell to a fully formed adult body containing numerous organs and tissues, >200 cell types and a total of about 50 trillion cells. One of the ways our cells turn genes “on and off” is through the addition or removal of acetyl moieties (CH3CO) to proteins called histones. Our DNA is stabilized and compacted by being wrapped around groups of 8 histones to form nucleosomes. The addition of acetyl groups to histones loosens that wrapping, allowing the DNA to be more accessible for transcription, “turning on” genes of interest. This process is coordinated by enzymes called histone acetyltransferases (HATs, also called lysine acetyltransferases) and histone deacetylases (HDACs or lysine deacetylases). The precise coordination of these enzyme activities is essential to allow our stem cell populations to replenish themselves or differentiate along different pathways. Many of these enzymes have been described as being key regulators for hematopoietic, brain, and mesenchymal stem cells. This review describes how HATs and HDACs regulate stem cell processes and what is currently known regarding the roles of acetylation of histones in stem cell biology.

**Abstract:**

Acetylation of histones is a key epigenetic modification involved in transcriptional regulation. The addition of acetyl groups to histone tails generally reduces histone-DNA interactions in the nucleosome leading to increased accessibility for transcription factors and core transcriptional machinery to bind their target sequences. There are approximately 30 histone acetyltransferases and their corresponding complexes, each of which affect the expression of a subset of genes. Because cell identity is determined by gene expression profile, it is unsurprising that the HATs responsible for inducing expression of these genes play a crucial role in determining cell fate. Here, we explore the role of HATs in the maintenance and differentiation of various stem cell types. Several HAT complexes have been characterized to play an important role in activating genes that allow stem cells to self-renew. Knockdown or loss of their activity leads to reduced expression and or differentiation while particular HATs drive differentiation towards specific cell fates. In this study we review functions of the HAT complexes active in pluripotent stem cells, hematopoietic stem cells, muscle satellite cells, mesenchymal stem cells, neural stem cells, and cancer stem cells.

## 1. Introduction

Somatic cell-types in the body have the same DNA sequences, yet each cell type possesses unique properties and functional capabilities. This is the result of different combinations of genes being expressed in each type [1]. Thus, the mechanisms that determine which genes get expressed and which are silenced are ultimately the regulators of cell identity. Stem cells, cells with the potential to differentiate into other cell types, are no exception. Epigenetic mechanisms control differential gene expression through the tightly regulated processes of transcriptional activation and repression. These mechanisms change the physical structure of chromatin but do not alter gene sequences. A variety of biochemical processes produce signals that are interpreted to allow cells to determine whether a gene eventually gets transcribed or not, and to what degree. For example, DNA and proteins can be biochemically modified—and depending on the modification, the gene will be turned on or off. One well defined example is DNA methylation, where cytosine nucleotides can have a methyl group attached to them [2]. At promoters, this typically is associated with gene repression [3]. Another category, which this review will mainly cover, is histone modification. Negatively-charged DNA within a nucleus is bound to positively-charged histone octamer complexes [4]. Depending on how tight the DNA is bound to the histone octamers, this regulates its accessibility to transcription factors and ultimately its ability to be expressed [5,6]. Tightly compacted DNA, referred to as closed chromatin or heterochromatin, is sterically hindered from transcriptional machinery binding; hence genes in these regions tend not to be expressed or expressed at low levels [7]. When the cell receives signals to express these genes, cell machinery is activated in order to loosen the DNA from the histones, allowing an open chromatin or euchromatin conformation to be achieved [8]. This can be accomplished by a variety of mechanisms. For example, a pioneer transcription factor may be able to recognize nearly inaccessible sequences to some degree and then recruit chromatin and histone modifiers or nucleosome remodelers to the target site [9]. Histones undergo numerous biochemical modifications, such as methylation, acetylation, phosphorylation, ubiquitylation, biotinylation, and sumoylation, as well as less well understood modifications like citrullination, deamination, GlcNAcylation, and others. Many of these modifications occur on the positively charged tails that protrude from the histone octamer; however, modifications can occur on the histone core too [10]. The function of the modification depends on the type of biochemical molecule placed on the histone along with which histone subunit and residue is modified [11]. Methylation is regulated by histone methyltransferases (HMT) which add a methyl group, and histone demethylases which remove methyl groups [12,13]. The methylation status can often be used to predict which genes are transcriptionally activated and which are repressed [14]. For example, a trimethylation of lysine 3 on histone 3 (H3K4me3), is typically associated with the promoters and enhancers of active genes [15]. Meanwhile, trimethylation of lysine 27 on histone 3 (H3K27me3) is usually associated with repressed genes [16]. Depending on the methylation mark, the mark can functionally recruit other complexes such as acetyltransferases or deacetylases that lead to gene activation or repression.

Histone acetylation commonly occurs on the lysine residues of histone tails; however, certain residues in the globular domain of histones can also be acetylated [17]. Acetylation neutralizes the positive charge on lysine causing the electrostatic association between the histones and the DNA to get weaker. By weakening this interaction, DNA becomes more accessible to transcription factors [18]. Therefore, the acetylation of histones is typically associated with the activation of gene expression. Furthermore, acetylation at certain histone residues can be recognized by certain protein domains, such as bromodomains. Similarly to histone methylation, this allows the recruitment of other factors to the DNA [19]. Acetylation is regulated by histone acetyltransferases (HAT) which add acetyl groups, and histone deacetylases (HDAC) which remove acetyl groups [20,21]. Many KATs target lysine residues and are not restricted to histones; hence, they are often named lysine acetyltransferases (KAT) or lysine deacetylases (KDAC). KATs can acetylate non-histone substrates such as proteins, which can regulate their activity or fate. The source of the acetyl group that KATs use comes from acetyl-CoA [22]. KDACs catalyze the removal of acetyl groups using NAD+ as a cofactor [23] and are frequently characterized to be zinc-dependent [24]. Since both acetyl-CoA and NAD+ are also products of cellular respiration, cell metabolism contributes to regulating the levels of acetylation and deacetylation [25]. KATs can be grouped into two main categories based on their localization: Type-A KATs are found in the nucleus while Type-B KATs are found in the cytosol [26]. KATs can also be grouped based on sequence homology, structure, and function. For example, the GCN5-related N-Acetyltransferase (GNAT) family generally have a bromodomain that allows them to bind to acetylated lysines [27]. The p300/CBP family consists of the two paralogs CREB-binding protein (CBP) and p300. Both of these contain a bromodomain and have conserved KAT domain sequences [28]. The MYST family is another group which typically is characterized by having zinc fingers and chromodomains. They also have a conserved MYST domain [27]. There are other KATs that do not belong to these families with other functions. Examples of some are those involved in RNA synthesis. TAFIIIC is a general transcription factor involved with RNA polymerase III that has KAT components within [29]. TAFII250 is a KAT and also a component of the TFIID subunit in the RNA polymerase II initiation complex [30]. In this review, we focus on the role of histone acetylation in the regulation of stem cell properties. Figure 1 shows a scheme of different KAT complexes and their main core components.

Stem cells are found in numerous tissues and have the ability to differentiate and self-renew [31]. Differentiation is the process where a stem cell divides and turns into well-defined cell types. The extent of different cell types a stem cell can differentiate into determines its “potency”. During development, the first cells that are capable of differentiating into all cell types, including the placenta and embryo, are deemed “totipotent” [32]. “Pluripotent” cells like embryonic stem cells are capable of differentiating into all of the cell types that compose the embryo [33]. Cells that are capable of differentiating into various cell types within a particular lineage are known as “multipotent”. Finally, stem cells capable of only differentiating into one cell type are called “unipotent”. Another property of stem cells is their ability to enter a state of quiescence, or reversible cell cycle arrest for long periods of time. This allows them to remain dormant and functional until they are signaled to divide again. Whether a cell commits to differentiate toward a certain program, or chooses to self-renew, is governed by the genes that are activated or deactivated through signaling within the stem cell niche or microenvironment. In eukaryotic cells, the activation of most genes is a tightly regulated, and chromatin-dependent process. KATs play a key role in gene activation, so characterizing the various KAT complexes and their mechanisms in transcriptional activation of stemness and differentiation genes is proving very informative for understanding the contributions of chromatin structure to stem cell maintenance and differentiation.

## 2. KAT Function in Pluripotent Stem Cells

During human development, pluripotent stem cells emerge as the inner cell mass contained in the blastocyst, known as embryonic stem cells (ESC). These cells give rise to all of the cell-types that compose an adult human; hence, they maintain a relatively unspecialized state. Another cell-type with pluripotency is induced pluripotent stem cells (iPSC). These cells are generated by reprogramming differentiated cells into a state similar to the ESCs. This was initially achieved by inducing the expression of the transcription factors OCT4, SOX2, c-Myc, and Klf4, known as the “Yamanaka Factors” [34]. This helped establish our understanding of the pluripotency transcription network in which transcription factors like OCT4, SOX2, and Nanog are critical in maintaining the pluripotent state. After years of developments in the field we now know that other transcription factors can function in stem cell pluripotency such as SALL4 [35]. These transcription factors not only activate expression of pluripotency genes but can also activate each other’s expression.

KAT complexes have been shown to target different pluripotency genes and, as a consequence, particular histone acetylation patterns have been found to occur frequently in human ESCs. For example, H3K56ac was shown to mark the promoters of OCT4, SOX2, and Nanog. A number of MYST family KATs are capable of keeping ESCs in an undifferentiated state. The Mof complex plays a key role in ESCs by participating in the expression of both Nanog and many Nanog target genes [36,37]. Deletion of Mof leads to aberrant expression of OCT4, SOX2, and Nanog, and, ultimately, loss of pluripotency [36]. Another member of the MYST family, MYST2/HBO1, functions in the acetylation of histones H3 and H4 in mESCs, as well as in other somatic cells [38]. MYST2/HBO1 has been shown to associate with the Nanog gene’s promoter and facilitate OCT4 binding [39]. The Tip60-p400 complex, another MYST family member also plays a function in maintain ESC fate. Tip60-p400 plays two parts: it can help activate genes related to cell cycle and cell proliferation; however, its main function is to repress genes that are involved in differentiation [40]. This repression seems to be facilitated by Tip60-p400′s acetylation of target histones [41]. This important example illustrates how KAT activity can be both positive and negative to gene transcription. Tip60-p400 is recruited to target promoters by the H3K4me3 mark. Furthermore, this recruitment seems to be facilitated by Nanog independent of H3K4me3 [42]. Other KAT families have been shown to play a role in maintaining ESC fate. The mammalian orthologue of the yeast GCN5, KAT2A, helps stabilize the pluripotency transcription network in mouse ESCs (mESC) by acetylating H3K9 at proximal regulatory regions of pluripotency genes, such as OCT4 and Nanog [43]. By inhibiting KAT2A, mESCs are induced to differentiate towards the mesendoderm lineage [43]. Another family that plays a part in pluripotency is the p300/CBP family. Both p300 and CBP have been shown to be recruited via direct interaction with Nanog and play a role in the gene activation of Nanog-targeted genes [44]. Furthermore, this gene activation is KAT-dependent and is critical for ESC self-renewal. The same study showed that the Nanog-p300/CBP complex plays a role in chromatin looping structures in ESCs. In contrast to helping maintain stemness, another well-documented change is the loss of multiple H4 acetylation marks as ESCs differentiate and lose their ability to be pluripotent [45].

## 3. KATs in Hematopoietic Stem Cells

Hematopoietic stem cells (HSC) generate all cell types found in the blood. They are able to differentiate into either myeloid progenitor cells or lymphoid progenitor cells. The myeloid lineage produces thrombocytes, erythrocytes, mast cells, basophils, neutrophils, eosinophils, and macrophages. The lymphoid lineage produces the natural killer T-cells, T-cells, and B-cells. Because they are restricted to only generating the different types of blood cells, they are classified as multipotent. HSCs are also one of the known populations of stem cells that are present in adulthood.

Within the MYST family of KATs, the MOZ complex has a major function in HSC maintenance. MOZ-deficient mice were shown to be embryonic lethal and had a significantly reduced level of HSCs in fetal liver [46]. Furthermore, MOZ was required for reconstitution of hematopoiesis for HSCs that were transplanted into mice [46,47]. MOZ has also been shown to promote HSC maintenance through an unusual mechanism. MOZ binds to the p16(INK4a) tumor suppressor promoter and inhibits transcription. This has been shown to help with HSC proliferation by preventing early senescence [48]. Suppression of a gene via a KAT is unusual and how MOZ does this has not been fully characterized. Another MYST family member, MORF, plays a role in HSC cell fate determination. Knockdown of MORF in mouse HSCs leads to increased differentiation towards the myeloid lineage and decreased erythropoietic activity. These results may contribute to explaining why age-associated decrease in MORF is associated with HSCs biasing towards production of myeloid cells [49]. Both p300 and CBP have been shown to help HSCs maintain their stemness through acetylation of genomic loci of HSC genes. As predicted from its role in maintaining stemness, pharmacological inhibition of p300 and CBP leads to differentiation of HSCs [6].

## 4. KATs in Muscle Satellite Cells

During muscle development, myoblasts proliferate and then commit to becoming myotubes by fusion producing multinucleated cells that form the muscle fibers. A subset of myoblasts de-differentiate to become quiescent satellite cells [50]. Satellite cells are the adult muscle stem cells that are responsible for muscle regeneration following injury. Upon receiving appropriate extracellular signals and their transduction within the cell, satellite cells can differentiate back into myoblasts [50]. A key factor that drives satellite cells to differentiate into myotubes is the master transcription factor MyoD. The KAT p300 plays a role in myotube differentiation by targeting MyoD gene regulatory elements, which leads to increased levels of MyoD expression [51]. In addition, p300 is capable of acetylating the MyoD protein itself, a post-translational mechanism that is crucial to the transcription factor’s activation [52]. Consistent with a role in promoting differentiation, H4 acetylation at MyoD-target genes does increase during differentiation [53] although genome-wide histone acetylation has been observed to be higher in satellite cells compared to when they differentiate into myotubes.

## 5. KATs in Mesenchymal Stem Cells

Mesenchymal stem cells (MSC) are multipotent cells capable of forming various cell types—primarily in the skeletal system. Human MSCs (hMSC) have been demonstrated to be able to differentiate into small subsets of cells that are found within multiple major embryonic lineages. Furthermore, hMSCs can be isolated from a variety of tissues in the body. Most hMSC cell lines reported in the literature are able to differentiate in vitro into osteocytes, chondrocytes, and adipocytes [54,55,56,57,58]. A large proportion of hMSCs can differentiate into pancreatic cells, neuronal cells, or hepatocytes. Some lineages are also able to form myocytes, cardiomyocytes, and melanocytes underlining their inherent plasticity [54,55,56,57,58]. The KAT GCN5, has been shown to drive osteogenic differentiation of hMSCs by inhibiting NF-κB via an acetyltransferase-independent mechanism [59]. GCN5 has a function in promoting MSC to cardiomyocyte differentiation too. This appears to occur by transcription factor ISL-1 bringing GCN5 to GATA4 and Nkx2.5 promoters leading to chromatin acetylation [60,61]. p300/PCAF KATs have also been shown to play a role in osteogenic differentiation in a BMP-mediated pathway [62]. An interesting note is that this is assisted by the SIRT6 histone deacetylase. Somewhat paradoxically, in a deacetylase-independent manner, SIRT6 seems to promote the targeting of p300/PCAF to osteogenic gene promoters where it can perform its KAT function [63].

## 6. KATs in Neural Stem Cells

Neural stem cells (NSC) are multipotent cells that differentiate into the different cell types of the nervous system, including neurons and glial cells. NSCs form the central nervous system during development where they begin within the neural tube. They give rise to neurons, then oligodendrocytes, and astrocytes in that order [64]. Neurogenesis has been shown to occur in adults as well, where NSCs have been found in certain regions of the brain [65].

CBP has been demonstrated to play roles in the proper differentiation of NSCs, neural precursor cell migration, and regulation of brain size. Loss of CBP leads to reduced neurogenesis and neuron density in certain regions of the mouse brain [66]. Loss of Brpf1, a component of many MYST family of KATs (such as MOZ/MORF and HBO1), leads to deregulation of NSC functions and neuronal migration in mice. Ultimately this leads to abnormal hippocampus morphology [67]. Hence, this implies that the MYST family has an importance in neural development. Consistent with the involvement of MYST KATs in neurogenesis, MORF helps establish olfactory bulb interneurons [68].

MORF has been shown to help maintain NSCs where its direct interaction with histone variant H3.3 promotes H4K16 acetylation. This helps NSCs proliferate and prevents premature differentiation during development [69]. Another KAT important for NSC self-renewal is MOZ. Like previously noted for HSCs, MOZ binds to the p16(INK4a) gene promoter and prevents its expression in NSCs [48]. MORF has also been shown to be is involved in the proliferation and maintenance of adult NSC populations [68]. High expression of MORF in NSCs in comparison to other cell types supports its function as a marker for NSC stemness [70].

Certain KATs have also been associated with promoting differentiation into the neural lineage. For example, the acetyltransferase KAT6B, found in MOZ/MORF complexes, helps with the efficient differentiation of ESCs into neural progenitor cells [71]. Another interesting note is the change in acetylation patterns observed during ESC commitment to the neural lineage—where H3K9ac levels decrease at pluripotency genes and increase at neural differentiation genes [72]. This study found that p300 depletion led to decreased expression of early neural lineage genes; hence, p300 appears to be involved in ESC to neural differentiation.

## 7. KATs in Cancer Stem Cells

While the properties of stem cell maintenance are critical to the healthy functioning of an organism during development and adulthood, the same mechanisms are deleterious to the hosts if adapted by tumorigenic cells. These “cancer stem cells” (CSC) have been found in various sources, including the blood as leukemia and within different solid tumors [73,74]. CSCs have properties of a stem cell in that they can undergo self-renewal and differentiate into other cell types. By differentiating into other types, this can make it difficult to treat or identify the original CSC cell type. Furthermore, by maintaining a “de-differentiated” state, CSCs can grow in a variety of niches in contrast to most differentiated cells which can only survive in specialized niches. This allows CSCs to spread and colonize other tissues in the body. CSCs can also enter a state of quiescence, which makes them particularly resistant to therapies that target dividing cells. Two main hypotheses exist to explain how CSCs form. One suggests that differentiated somatic cells obtain malignant and stem-cell like characteristics through mutation or epigenetic modifications. These could allow the cell to acquire a de-regulated cell cycle, possibly becoming replicatively immortal, and then de-differentiate over time. This could occur if these cells underwent an epithelial-mesenchymal transition (EMT) in which they pass through a less differentiated state. The second explanation suggests that a normal stem cell directly gains the ability to become tumorigenic due to mutation or epigenetic change. This could occur during adulthood or during development as the cell acquires changes that promote tumorigenesis. Due to the heterogenous nature of cancer cells, it may be hard to pinpoint common gene regulatory mechanisms for stemness in a single tumor, let alone different types of CSCs. Indeed, several single cell sequencing studies of tumors have suggested that a typical tumor may consist of different, distinct types of CSCs that are plastic in nature [75,76]. Nonetheless, here we will discuss the nature and types of KATs found to be dysregulated in certain CSC types. The identification of KAT complexes critical for CSC maintenance would logically lead to the pursuit of drug inhibitors to target these KATs. Thus, we also mention the small molecules that have been found to destabilize CSC maintenance in this section.

CSCs were first found in acute myeloid leukemia (AML) as leukemia stem cells (LSC) [73]. The MYST family member HBO1 was later found to be key for LSC maintenance [77]. HBO1 acetylates H3K14 which helps maintain the expression of Hoxa9 and Hoxa10. The Hoxa cluster of genes is critical for development of the hematopoietic lineage and maintains the differentiation potential of HSCs. Hox9 has also been characterized as a key driver of leukemogenesis. Another pair of MYST KATs found to play a role in leukemogenesis are MOZ and MORF. This does not come as too much of a surprise as we have noted the critical function of MOZ/MORF in HSCs. During leukemogenesis, the endogenous MOZ and MORF genes have been reported to be disrupted, leading to changes in functional activity. For example, the MOZ gene can undergo a chromosome translocation and fuse with another KAT from another family, CBP. These KAT fusions then lead to dysregulated chromatin acetylation. Another MOZ fusion identified is the MOZ-TIF2 product, where TIF2 is an interactor of the p300 KAT. This MOZ-TIF2 fusion has been shown to transform mouse hematopoietic progenitors into LSCs and promote their maintenance [78].

Due to the effects of KATs on stemness in cancer, this has prompted the development of drug inhibitors as candidates for chemotherapy. An example is the previously mentioned Hoxa9/Hoxa10 study where these authors chemically inhibited the acetyl-coA binding site of HBO1 with a small molecule they generated called WM-3835 [77]. This ultimately led to decreased LSC growth. 3-methylcyclopentylidene-[4-(4′-chlorophenyl) thiazol-2-yl] hydrazone (CPTH6) is a PCAF and GCN5 KAT inhibitor that targets the growth of lung cancer stem-like cells (LCSC) [79]. CPTH6 exerts an oncolytic effect by acting through apoptosis and autophagy pathways. Additionally, it induces loss of CSC markers such as CD133 and ALDH. The inhibition of some KAT complexes has been found to be cytotoxic to cancer cells in general. For example, inhibition of PCAF with 2-acylamino-1-(3- or 4-carboxyphenyl) benzamides is cytotoxic to several human carcinomas such as colon, breast, lung, hepatoma, cervical, and kidney [80]. WM-8014 and WM-1119 were developed to inhibit the acetyltransferase subunit of the MOZ complex, KAT6A. Treatment with WM-8014 and WM-1119 inhibited lymphoma proliferation and induced senescence via the p16INK4A and p19ARF pathway [81]. Currently, there is a lack of KAT inhibitors in clinical trials, unlike inhibitors for the HDACs, which are involved in the opposing process of deacetylating histones [82]. In Table 1, we reference some of the known KAT inhibitors and their activities in human cells.

## 8. Summary and Conclusions

Characterizing the various KAT complexes and the genes they help regulate will help us understand the complex transcriptional network that governs each type of stem cell and differentiation fate. The current literature has shown that depending on the stem cell type, different KATs from each of the families have varying importance in self-renewal and differentiation properties. However, one family of KATs that seems to be commonly implicated in affecting various stem cell types is the MYST family that regulate transcription of various genes related to cell fate determination. Furthermore, HBO1 is known to function at the DNA replication fork, thus affecting cell replication. Whether this suggest that the MYST family has a specific function in regulating transcription of genes essential for stem cell maintenance and differentiation or whether they regulate transcription in a more global fashion remains to be determined.

From a mechanistic point of view, beyond identifying the genes KATs can regulate, it is important to understand how they regulate them. KAT function is not as simple as acetylating histones to allow transcriptional machinery to access the DNA. As this review has highlighted, there are various examples of non-canonical KAT mechanisms. In the example of Tip60-p400, the acetylation of chromatin ultimately leads to the repression of the target genes. Other reports suggesting KAT-dependent gene silencing may involve recruitment of other factors independent of acetyltransferase activity. Understanding these exact mechanisms would be critical for designing pharmacological strategies to target them. For example, if you wanted to inhibit the function of a KAT, you would have to determine whether to target a protein interacting domain or the acetyltransferase domain itself. In addition to KAT inhibitors, there are also a small number of KAT activators that have been described. Some examples would be the CTPB compound (a p300 activator) [83], LoCAM (a PCAF activator but a p300 inhibitor) [108] and CSP-TTK21 (another p300 activator). CSP-TTK21 has shown the exciting capability of reversing key symptoms and characteristics associated with Alzheimer’s disease lesions in mice [109], perhaps by inducing axon regeneration. However, since many more KAT inhibitors and their activities are much better described and characterized for their various biochemical and physiological activities, we have focused on them in this review.

Needless to say, from a theoretical perspective, understanding stem cell fate decisions will take much more than just characterizing KAT function in each stem cell type. Each KAT found in the major KAT families (GCN5, MYST, and p300/CBP), does not function exclusively in a single cell type. KATs primarily function as a tool for gene activation. To understand how this tool is functionally coordinated, we must understand the mechanisms that recruit them to a specific target gene. One known mechanism of KAT recruitment to genes involves a histone reader binding to a certain methylation pattern on a histone and then it proceeding to recruit its respective KAT complex(s). For example, the inhibitor of growth 5 (ING5), is a histone reader that recognizes the H3K4me3 mark of active transcription and is a part of various MYST family KATs. Some studies have suggested the importance of ING5 in maintaining stemness in various stem cell types [110], including in cancer stem cells [111]. This is in agreement with literature supporting the function of MYST KATs in maintaining stem cells [36,37,38,46,47,48,49,69]. However, the recognition of histone marks does not explain how KATs are specifically brought to specific genes and corresponding target sequences. Sequence-specific targeting of KATs to various genes remains an enigmatic area to date and will likely be determined to be a function of interaction between functional domains of KAT complex members and sequence-specific DNA binding proteins. Additional studies suggest that lncRNAs that bind to genes in a sequence-specific manner may also serve to help recruit KAT complexes [112]. Thus, each gene may have its own particular mechanism for KAT recruitment and so it may prove to be the case that each gene may have to be examined individually to fully understand how transcription is fine-tuned to regulate stem cell quiescence, renewal, and differentiation.

## Figures and Tables

**Figure 1 cancers-13-02407-f001:**
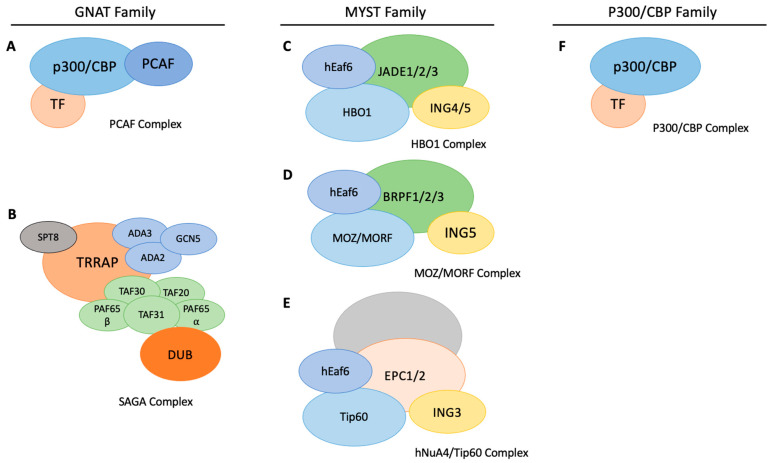
Examples of known KAT complexes from various KAT families. (**A**) The PCAF complex is composed of a DNA-binding transcription factor, p300/CBP, and bound to it is the GNAT-family KAT, PCAF. (**B**) The SAGA complex belongs to the GNAT family and uses GCN5 as the histone acetyltransferase. (**C**) The HBO1 complex is composed of the HBO1 histone acetyltransferase, ING4/5 as the histone-binding subunit, and JADE1/2/3 for scaffolding. (**D**) The MOZ/MORF complex is composed of MOZ/MORF acetyltransferase, BRPF1/2/3 for scaffolding, and ING5 for binding histones. (**E**) The hNuA4/Tip60 complex is composed of the Tip60 acetyltransferase, ING3 histone-binding subunit, and scaffolding subunit EPC1/2. The unlabeled structure behind EPC1/2 refers to variable components that are known to change depending on biological/cellular context. (**F**) p300/CBP complex is composed of a transcription factor that acts as a DNA-binding element and brings p300/CBP acetyltransferase to the chromatin.

**Table 1 cancers-13-02407-t001:** List of KAT inhibitors with known effects on human cells.

Name on Original Publication	Inhibited Complex	Biological Effects Noted	Reference
Anacardic acid	p300 PCAF	Targeted tyrosinase, tissue factor VIIa, xanthine oxidase, phosphatidylinositol-specific phospholipase C, lipoxygenase, and cyclooxygenase	[83,84,85,86,87,88,89]
MG149	Tip60 MOF p300/PCAF	Inhibited the expression of pro-inflammatory genes	[90,91,92]
2-acylamino-1-(3- or 4-carboxyphenyl)benzamides (small molecule inhibitors)	PCAF	Cytotoxic to several cancer cell lines	[80]
Curcumin	p300/CBP	Was capable of alleviating ventricular hypertrophy, microangiopathy and heart failure	[93,94,95,96]
Garcinol	P300 PCAF	Takes part in the response to oxidative-stress inflammatory and apoptosis processes, regulation of early growth response protein 1 (EGR-1), proliferation, metastasis and angiogenesis, and in addition epigenetic pathways	[83,97,98,99,100,101,102]
LTK-14	P300	Inhibited the multiplication of the HIV virus while being nontoxic for T-cells.	[103]
Pro-B3	p300 60% inhibition Tip60/PCAF/CBP < 40% inhibition	Inhibition of the activation of the androgen receptor (AR) by p300-mediated both in vitro and in vivo; resulted in the suppression of prostate cancer cell growth	[104]
Delphinidin	P300	The inhibition of expression of inflammatory cytokines in MH7A cells along with the release of cytokines in Jurkat T lymphocyte cell line.	[105]
NK13650A	P300	Supressed the transcription of the AR and slowed the growth of prostate cancer cells	[106]
WM-8014	KAT6A	Inhibited cell proliferation and induced senescence	[81]
TH1834	Tip60	Induced DNA damage and cell cycle dysregulation into cancer cells.	[107]

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
