# Peer review of "Histone Acetyltransferases and Stem Cell Identity"

_cancers, 2021, doi:10.3390/cancers13102407_

Round 1

Reviewer 1 Report

This study demonstrates the important roles of HATs. Figure 1 may be revised to indicate a component behind EPC1/2 in Figure 1E. The differences between KAT and HAT might be clearer in the legend for Figure 1.

Author Response

We thank the reviewer for positive comments and the suggestions to clarify HAT vs KAT terminology and explain why one component of the EPC1/2 complex was not labelled, which we have done.

Reviewer 2 Report

The review article of He et al., discusses the role of acetylation of histones as a key epigenetic modification involved in transcriptional regulation. The review is nicely written and the efforts taken by the authors is appreciated. 

I recommend acceptance of this article in its present form after grammar checks and few other English language editing. 

Author Response

We have rechecked the grammar and spelling as suggested.

Reviewer 3 Report

The review article by He and colleagues on the role of histone acetyl transferases (HATs) in stem cell biology is very well written, clearly structured and informative. I only have a small number of minor points which should be addressed by the authors:

-Line 45: The term “tightly bound DNA” might be misleading. I guess the authors mean “tightly compacted DNA”?

-Line 69/70: The authors correctly describe the acetylation-mediated interactions between DNA and histones as charge-driven. However, when talking about transcriptional regulation in a broader context, also other aspects of histone acetylation come into play, such as e.g. the recruitment of acetyl-reading proteins. The authors should at least mention this point.

-Line 103: too much spacing between the words “stem cells…….are capable”.

-References are sometimes not within the sentence where they belong. Example: Line128: …such as “…SALL4. [34] These…”. Other examples: Lines 209, 273, 280.

-Table1 lists HAT inhibitors and their impact on stem cell biology. What about available HAT activators? Which ones are available? What is their impact on stem cells? Could they be used in e.g. regenerative medicine?

Author Response

We thank the reviewer for the suggestions and agree with all of them and so we have:

1) changed terminology from "tightly bound DNA" to "tightly compacted DNA" 2) we note that certain histone marks like methylation can result in recruitment of other epigenetic regulators like acetytransferases/deacetylases and now mention that acetylation marks can be recognized by proteins with bromodomains.

3) we removed excess space

4) we fixed the placement of periods after references

5) we have mentioned that some HAT activators have been described and note that one has shown some efficacy in treating a mouse model of Alzheimer's disease but since there are few activators described we remain focussed on the much better described functions of HAT inhibitors.